# The design and implementation of restraint devices for the injection of pathogenic microorganisms into *Galleria mellonella*

**Lance R. Fredericks**[1], **Mark D. Lee**[1☯], **Cooper R. Roslund**[1☯], **Angela M. Crabtree**[1], **Peter B. Allen**[2], **Paul A. Rowley**[1]*

1 Department of Biological Sciences, University of Idaho, Moscow, ID, United States of America,
2 Department of Chemistry, University of Idaho, Moscow, ID, United States of America

☯ These authors contributed equally to this work.
* prowley@uidaho.edu

**Data Availability Statement:** The design schematics for an acrylic injection chamber have been deposited in the Thingiverse public repository

## Abstract

The injection of laboratory animals with pathogenic microorganisms poses a significant safety risk because of the potential for injury by accidental needlestick. This is especially true for researchers using invertebrate models of disease due to the required precision and accuracy of the injection. The restraint of the greater wax moth larvae (*Galleria mellonella*) is often achieved by grasping a larva firmly between finger and thumb. Needle resistant gloves or forceps can be used to reduce the risk of a needlestick but can result in animal injury, a loss of throughput, and inconsistencies in experimental data. Restraint devices are commonly used for the manipulation of small mammals, and in this manuscript, we describe the construction of two devices that can be used to entrap and restrain *G. mellonella* larvae prior to injection with pathogenic microbes. These devices reduce the manual handling of larvae and provide an engineering control to protect against accidental needlestick injury while maintaining a high rate of injection.

## Introduction

The larvae of the greater wax moth *Galleria mellonella* is an important animal model for studying host-pathogen interactions and for the discovery of novel antimicrobial therapeutics. The popularity of this model organism is driven by the low cost of purchase and the reduced ethical concerns for the experimental manipulation of insects. This allows the challenge of a large number of larvae in a single experiment, which can improve the statistical power of an assay. Starting in the 1940s, a diversity of viral, bacterial, fungal, and nematode pathogens, have been studied for their ability to cause disease in *G. mellonella* larvae [1–15]. Importantly, *G. mellonella* can be maintained at mammalian body temperature and the outcomes of infection can reproduce that of mammalian animal models [16–18]. This is likely due to similarities in the innate immune response to pathogens mediated by elements of cellular and humoral immunity between insects and mammals [19–21]. *G. mellonella* has also been used extensively for compound toxicity screening [22]. There are several methods to introduce compounds and

(https://www.thingiverse.com/), identifier number 4170068, LarvaWormCorralV4.

**Funding:** The research was supported by funds provided to PAR by the Institute for Modeling Collaboration and Innovation at the University of Idaho (NIH grant P20 GM104420), the Institutional Development Award (IDeA) from the National Institute of General Medical Sciences of the National Institutes of Health under Grant #P20GM103408, the National Science Foundation Science and Technology Center on evolution in action, DBI-0939454 and the National Science Foundation grant number 1818368.Funding was also provided by the Office of Undergraduate Research at the University of Idaho awarded to LRF and CRR. Publication of this article was funded by the University of Idaho - Open Access Publishing Fund. The funders had no role in study design, data collection and analysis, decision to publish, or preparation of the manuscript and any opinions, findings, and conclusions or recommendations expressed in this material are those of the author (s) and do not necessarily reflect the views of the funders.

**Competing interests:** The authors have declared that no competing interests exist.

pathogenic microorganisms into *G. mellonella*, including topical application, feeding, baiting, oral gavage, submersion, and direct injection into the hemocoel [9,11,23–26]. The latter method is often favored because of the ability to control the dosage and timing of injections.

Despite the many benefits, there are challenges with using *G. mellonella* larvae as an animal model, most notably standardizing the health and developmental stage of the larvae. This is especially difficult when larvae are purchased from commercial sources that are primarily focused on providing feed and bait for the pet and angling communities [27,28], although there are commercial pipelines for scientific-grade larvae [29]. Another difficulty is the manipulation and restraint of small larvae during experimental injections. These experiments require both biological containment and the adequate protection of personnel from needlestick injuries and laboratory-acquired infections. The most basic technique of larval injection calls for the restraint of a larva between finger and thumb during the injection process [9,11,14,30]. With the operator's hands protected by latex gloves, this method offers maximum dexterity. However, the close proximity of a pathogen-filled needle to inadequately protected fingers presents a significant biological safety hazard that exposes personnel to a high risk of accidental needlestick injury. This particular restraint procedure is in conflict with biosafety guidelines for the implementation of policies for improved work practices that minimize needlestick injuries whenever possible [31]. Alternatively, the safe handling of *G. mellonella* larvae can be achieved with the use of needle resistant gloves or forceps, but with a loss of manual dexterity and the potential to cause animal injury or stress that can alter pathogen susceptibility [32]. Furthermore, needle-resistant gloves are made of porous materials that require covering with disposable laboratory gloves to prevent biological contamination, which further limits manual dexterity. To maximize safety, humane physical restraint devices have been used routinely for the handling and manipulation of laboratory animals [33]. For the injection of *G. mellonella*, there is also a device named the "Galleria grabber" that has been developed for the restraint of larvae between layers of a sponge [34]. This method enables injection without the need for grasping larvae between finger and thumb and offers the user protection from accidental needlestick injury. However, the use of a porous sponge for multiple injections increases the chance of its contamination by pathogenic microbes, which presents a challenge for effective decontamination.

In this study we present two simple restraint devices that can be fabricated from micropipette tips or acrylic glass (also known as poly(methyl methacrylate) or Plexiglass). These devices are easy to assemble and can be used to restrain large numbers of *G. mellonella* larvae in preparation for injection. The described protocol reduces the manual handling of larvae, enables a rapid injection speed, and allows the effective decontamination and sterilization of the devices for reuse. Both devices provide increased protection of the operator from accidental needlestick injury and laboratory-acquired infection.

## Materials and methods

### Culturing and preparation of yeast cells for injection

*C. glabrata* ATCC2001 was maintained using yeast extract peptone dextrose growth media (YPD). Several days prior to injection, *C. glabrata* was streaked out to clones from a -80˚C frozen stock in YPD with 15% glycerol. A single colony of yeast cells was grown overnight at room temperature to stationary phase in a 2 mL culture of liquid YPD medium. Stationary phase cultures were diluted 1/20 into a 125 mL flask and grown at room temperature until an $OD_{600}$ of 1.5 was reached. Hemocytometer counts of these cultures were used to determine the number of yeasts used for each injection ($8.0 \times 10^5$, $3.0 \times 10^6$, $4.6 \times 10^6$, and $5.0 \times 10^6$ *C.*

*glabrata* cells per injection). Prior to injection, yeast cells were harvested by centrifugation at $8,000 \times g$ for 1 min (25˚C) and suspended in filter sterilized PBS (pH 7).

### *G. mellonella* larva handling, care, and disposal

The described experiments followed the ARRIVE guidelines for reporting the use of animals in research [35]. Larvae were ordered from www.premiumcrickets.com using the "weather protect" service to maintain the temperature of the larvae during overnight shipping. Upon arrival, larvae were stored without light in wood shavings at 17˚C and were allowed to acclimatize for at least 2 days to control for the adverse physiological consequences of shipping [32]. *G. mellonella* larvae were used within 1 week due to the known physiological consequences of long-term storage [36]. Healthy *G. mellonella* larvae were selected by weight (175–225 mg), uniformity in color (little to no melanization), and responsiveness to touch. Prior to injection, larvae were incubated at 37˚C for 16 hours to allow for acclimatization to the assay temperature. Dead or unhealthy larva that are observed after the pre-incubation period were removed from the study prior to injection. Larvae exposed to pathogenic microorganisms were disposed of by placing them in secondary containment and incubating at -20˚C for 24 hours before sterilization by autoclaving.

### Fabrication of restraint devices for *G. mellonella*

The primary consumable used for the creation of the restrain device was the 250 μL VistaLab micropipette tip (catalog number: 4058–2000). Other brands of pipette tips have been tested for their compatibility with this method (S1 Table). Micropipette tips were cut at predefined points to enable assembly (Fig 1). To construct the restraint device fabricated from acrylic glass, transparent clear acrylic glass was purchased from AliExpress (1 mm × 100 mm × 100 mm) and was cut to the desired shape with a $CO_2$ laser cutter (BOSS Laser LS-1416) using software provided by the manufacturer (available online in SVG format at https://www.thingiverse.com/ design ID 4170068, "LarvaWormCorralV4").

### Injection of *G. mellonella* larvae

Injections were performed with Hamilton 700 syringes (Model 701 N, Volume: 10 μL, Point Style: 2, Gauge: 26s and Model 1750 LTSN SYR, Volume: 500 μL, Point style: 4 Gauge: 26s) with a repeating dispenser used for multiple injections (Hamilton PB600-1). We also expect that this method is fully compatible with the use of insulin syringes with shorter needles as prolegs are almost always positioned close to the opening in the restraint devices. During injection, fingers were protected by a HexArmor PointGuardⓇ Ultra 4041 glove (Performance Fabrics Incorporated).

### Data analysis

The graphical representation of the average survival rates of *G. mellonella* larvae after injection and the Kaplan-Meier log-rank analysis were performed using R (version 1.1.419) with the packages "ggplot2", "dplyr", "survival", and "survminer". $LT_{50}$ was calculated using the package "MASS". A power analysis was performed to assess the required sample size using G*Power (V3.1) (One-tailed T-Test, $\alpha = 0.05$, $\beta = 0.2$, effect size = 0.8).

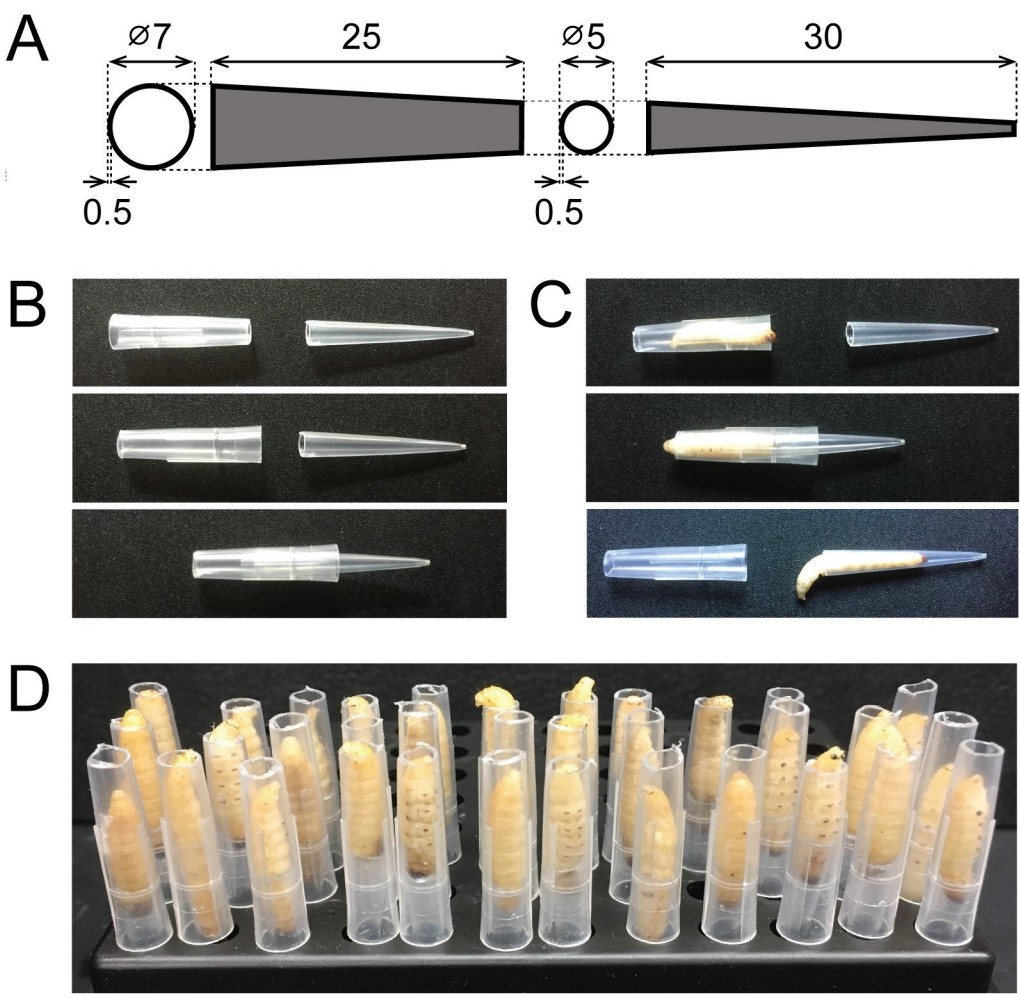

**Fig 1. Entrapment of *G. mellonella* in a restraint device constructed from a micropipette tip.** (A) Dimensions of a 250 μL VistaLab pipette tip indicating the cut site to create two 5 mm openings. (B) Assembly of the restraint device without a larva. (C) Top: Assembly of the restraint device with a larva captured abdomen-first into the larger half of the micropipette tip. Middle: A larva is entrapped by enclosing the chamber. Bottom: A larva is released by opening the chamber. Measurements are presented in millimeters. (D) Storage of multiple larvae in restraint devices prior to injection.

## Results and discussion

### The restraint of *Galleria mellonella* larvae using reusable restraint devices

A central challenge to the manipulation and injection of *Galleria mellonella* larvae is the ability to restrain them without injury prior to injection. We have previously observed that manual handling and restraint of larvae by inexperienced laboratory personnel can often cause animal injury and death. To minimize these undesirable consequences, we have designed and tested two types of devices for the restraint of individual *G. mellonella* larvae. These chambers can be constructed either from a disposable 250 μL or 1,000 μL micropipette tip of the required dimensions (Fig 1A and S1 Table) or assembled from laser cut acrylic glass (Fig 2). Both devices are designed to be fully reusable. To make the micropipette tip device, a tip is cut, and the two halves are used to entrap a larva with minimal handling (S1 Movie). Specifically, a larva is captured within the wider end of the cut pipette tip and entrapped by inserting the

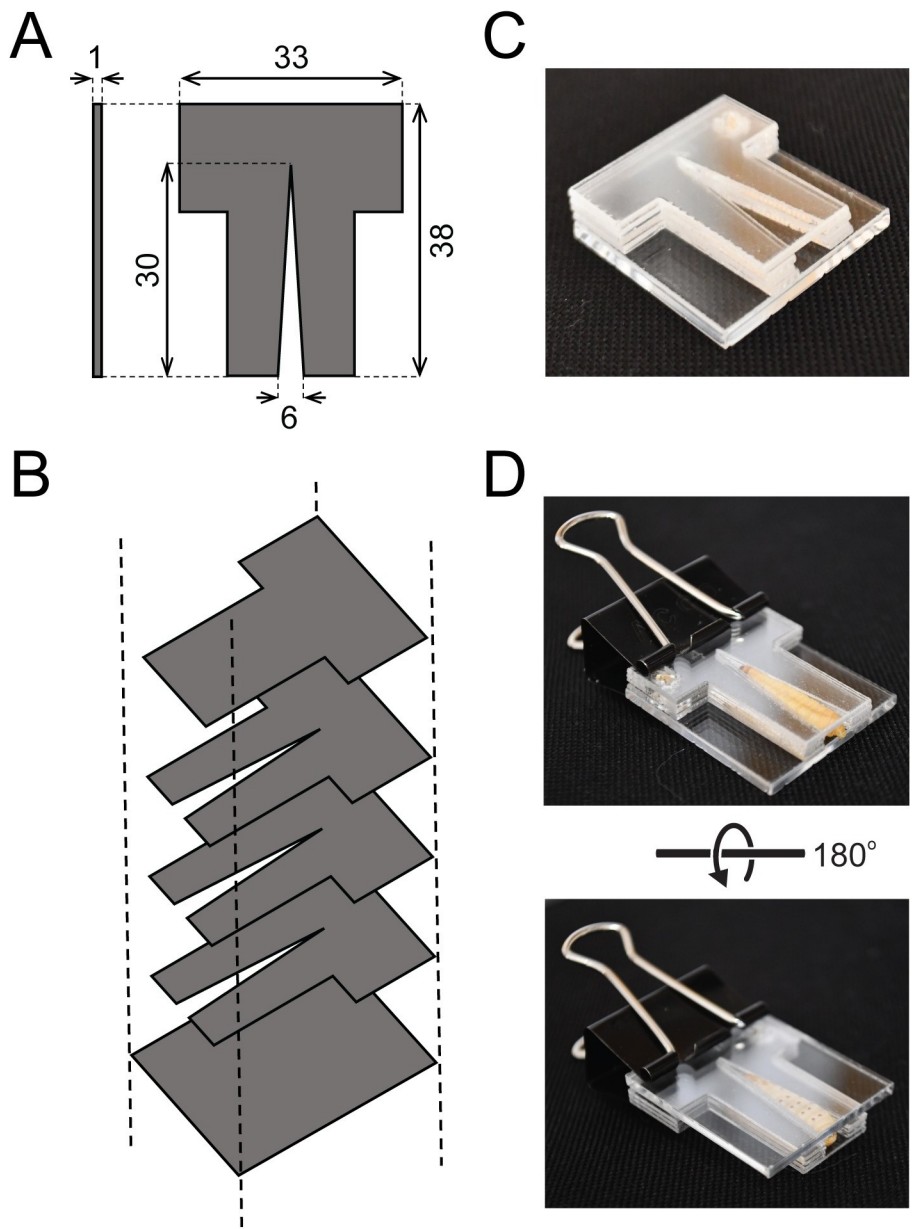

**Fig 2. Entrapment of *G. mellonella* in a restraint device constructed from laser cut acrylic glass.** (A) Dimensions of the acrylic glass restraint device. Measurements are presented in millimeters. (B) Schematic overview of the assembly of acrylic glass without a larva. (C) Assembly of the restraint device without a larva (D) A restraint device with larva captured head-first into the chamber. The ventral side of the larva is visible upon rotation of the chamber.

second half of the pipette tip to seal the restraint device at one end (Fig 1B). This procedure takes an average of 7 seconds (SD = +/- 3 seconds, n = 30). After entrapment, the escape time of larvae from the device was measured under ambient light conditions. After 30 and 60 minutes, 7% and 20% of larvae were observed to have exited the device, respectively (n = 30). This long occupancy time allows the loading of multiple larvae before injection and allows easier injection due to their predictable positioning within the device (Figs 1 and 2). Moreover, once entrapped, larvae generally wedge themselves into the device and remain motionless, even when turned to reveal their ventral side, which is likely due to their known aversion to light

(Figs 1D and 2D) [37]. Larvae that are attempting to exit the device prior to injection can be persuaded to re-enter with a gentle touch to their abdomen. After injection, a larva can be released by gently pulling the two halves of the device apart to allow egress (Fig 1C). To assemble the acrylic glass restraint device, the layers are stacked to construct a chamber, using a binder clip to secure them together (Fig 2). Depending on the size of the larvae, the height of the chamber can be adjusted by removing or adding notched layers. Importantly, a chamber that is too large will enable a larva to turn in the device and not present its ventral side for injection. Larvae are coaxed into entering the chamber head-first by gently pushing their abdomen towards the chamber opening. This procedure takes an average of 12 seconds (SD = +/- 5 seconds, n = 30).

## The injection of restrained *Galleria mellonella* larvae

To test whether the fabricated restraint devices are suitable for the injection of *G. mellonella*, larvae were first restrained using the described devices. Each larva was then injected with either phosphate-buffered saline (PBS) or the opportunistic fungal pathogen *C. glabrata* and the degree of insect mortality was measured. The relative performance of these devices was measured compared to larval restraint by finger and thumb using PBS. Injections were performed with Hamilton 700 syringes. Prior to injection, 70% ethanol was used to wash the syringe three times followed by a single wash with sterile distilled water before loading with PBS or *C. glabrata*. Fingers holding the restrain device or a larva were protected during injection with a needle resistant glove covered with a disposable laboratory glove to prevent contamination. To inject a larva immobilized within a restraint device, the needle is inserted through the wide end of the restraint device and used to pin down the larva at the last left proleg before puncturing the cuticle (Fig 3 and S2 Movie). The needle is inserted along the long axis of the larva at a shallow angle of 10–20˚ beneath the cuticle to avoid puncturing the midgut. The needle penetrates to a depth of <5 mm, whereupon the plunger is depressed to eject the contents of the syringe into the hemocoel. The shallow angle and depth of injection can be verified as the needle is visible through the cuticle. The injection time with a single channel

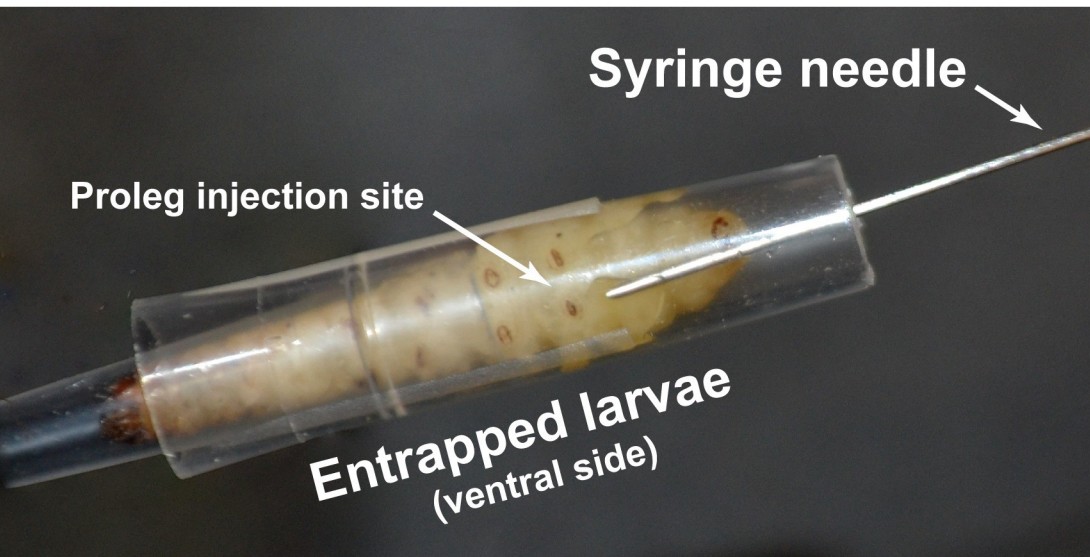

**Fig 3. A ventral view of a *G. mellonella* larva restrained within a micropipette device during the injection of *C. glabrata* into the last proleg.** The injection point in the last proleg of the larvae is punctured by a Hamilton syringe needle.

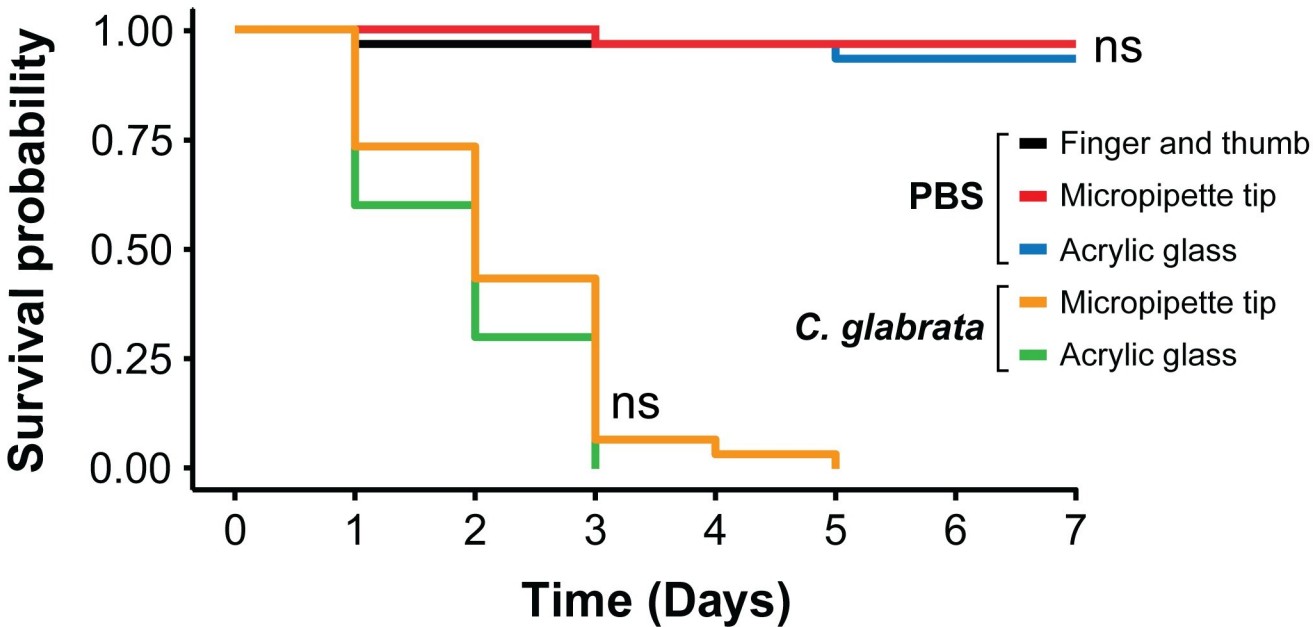

**Fig 4. Survival rates of *G. mellonella* larvae over seven days post-injection with PBS or *C. glabrata* cells after using different methods of larval restraint.** Statistical significance was judged by the Kaplan-Meier log rank test and compared three restraint methods after injection with PBS and the restraint devices after injection with *C. glabrata* (ns = not significant). Results presented for PBS injections were from a single biological replicate (n = 30). Results presented for *C. glabrata* injections were from two independent biological replicate (n = 15 for each replicate).

Hamilton syringe takes an average of 62 seconds per larva (SD +/- 12 seconds, n = 11) for the micropipette device, which includes loading the syringe, injection, larval release, and decontamination of the needle and syringe. Using a repeat dispenser increases the rate of injection to 8 seconds per larva for both the micropipette tip device (SD +/- 2 seconds, n = 30) and the acrylic glass device (SD +/- 3 seconds, n = 30). This is compared to 30 seconds (SD +/- 16 seconds, n = 30) to restrain and inject larvae between finger and thumb, using a needle resistant glove. After the withdrawal of the needle, the larva is placed in a 9 cm petri dish fitted with two 9 cm diameter disks of paper towel in the dark to promote the egress from the restraint device. Restraint devices can be easily decontaminated with 70% ethanol or 10% bleach after every use. While it is possible to repeatedly sterilize the micropipette devices using an autoclave, this method is not recommended for acrylic glass [37].

Once injected, larvae were observed for up to 7 days to assess the physiological consequences of the injection of $4.6 \times 10^6$ *C. glabrata* cells or PBS (Fig 4). Of the 90 larvae injected with PBS using different restraint methods, 96% of larvae survived injection (86/90) after 7 days and with no significant difference in survival of the larvae restrained by any method (Fig 4 and Table 1). There was also no significant difference in the survival curves of the larvae that were injected with *C. glabrata* (Fig 4 and Table 2), with an average time to 50% lethality ($LT_{50}$) of 1.8 and 1.4 days for the micropipette tip and acrylic glass devices, respectively (Table 2). In

**Table 1. A comparison of larval mortality using restrain devices compared to restraint by finger and thumb after the injection of PBS.**

| Device | n | Log rank test (P value) | Device loading time (seconds) | Injection time (seconds) |
|---|---|---|---|---|
| Finger | 30 | - | - | 30 |
| Micropipette tip | 30 | 0.56 | 7 | 8 |
| Acrylic glass | 30 | 0.99 | 12 | 8 |

**Table 2. A comparison of larval mortality using two types of restraint device during the injection of *C. glabrata*.**

| Device | n | Log rank test (P value) | LT50 (days) |
|---|---|---|---|
| Micropipette tip | 30 | 0.14 | 1.8 |
| Acrylic glass | 30 | | 1.4 |

an independent experiment, there was also a dose-dependent lethality with the injection of different numbers of *C. glabrata* ($8 \times 10^5$, $3 \times 10^6$, and $5 \times 10^6$) using the micropipette tip device (Fig 5). The $LT_{50}$ was calculated as 2.7 and 1.7 days upon injection of $3 \times 10^6$, and $5 \times 10^6$ *C. glabrata* cells, respectively, which was significantly different from the PBS control ($p < 0.0001$) (Fig 5). These data are comparable to previous studies that used alternative injection methods to infect *G. mellonella* with *C. glabrata* ATCC 2001 [7].

## Conclusion

The method described in this report uses engineering controls to spatially separate an injection needle from the fingers of laboratory personnel. When combined with a needle resistant glove, this almost completely eliminates the risk of needle injury and the accidental infection of laboratory personnel with pathogenic microbes. The devices that we have described to restrain *G. mellonella* larvae are inexpensive, non-porous, and can be reused multiple times after sterilization or decontamination. We have found that the ability to entrap multiple larvae enables a high rate of injection when using a repeat dispenser with a Hamilton syringe. The described devices are most effective with two people working together, one loading larvae into the restraint devices and the other performing injections. We find that the rate of injection is comparable to the Galleria grabber (20–25 seconds per injection) [34] and is close to the traditional methods of immobilization between finger and thumb protected by a needle resistant glove

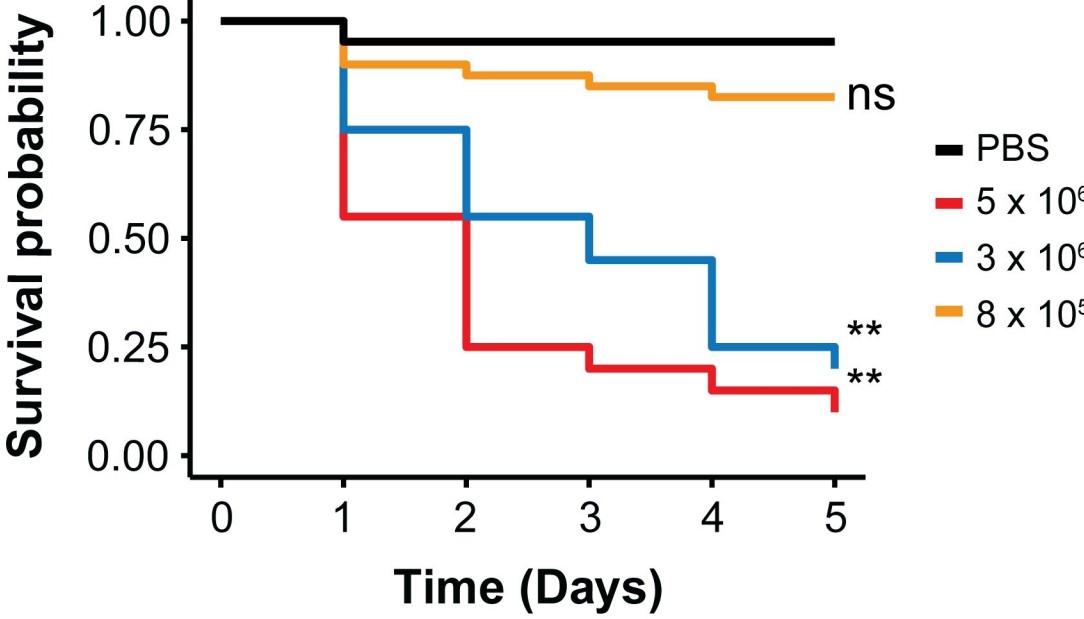

**Fig 5. Survival rates of *G. mellonella* larvae over five days post-injection with three different inocula of *C. glabrata* cells or with PBS.** Statistical significance relative to PBS was judged by the Kaplan-Meier log rank test (** $p < 0.0001$, ns = not significant). Results presented for these injections were from at least two independent biological replicates.

(30 seconds). The described method for the restraint of *G. mellonella* larvae offers a rapid and reproducible workflow for the injection of hazardous microorganisms.

## Supporting information

**S1 Movie. Entrapment of a *G. mellonella* larva in an injection device constructed from a micropipette tip.**
(MOV)

**S2 Movie. Injection of a restrained *G. mellonella* larva in the last proleg with a Hamilton syringe.**
(MOV)

**S1 Table. Brand compatibility with construction of the micropipette tip restraint device.**
(DOCX)

## Acknowledgments

We would like to thank the University of Idaho Biological Safety Officer Megan Grennille, and Nathan Taggart for critical reading of this manuscript. We would also like to acknowledge the contribution of Madison Chapman for technical assistance in the sorting and husbandry of *G. mellonella* larvae.

## Author Contributions

**Conceptualization:** Lance R. Fredericks, Cooper R. Roslund, Peter B. Allen.

**Data curation:** Angela M. Crabtree, Paul A. Rowley.

**Formal analysis:** Paul A. Rowley.

**Funding acquisition:** Paul A. Rowley.

**Investigation:** Lance R. Fredericks, Mark D. Lee, Cooper R. Roslund, Peter B. Allen.

**Methodology:** Lance R. Fredericks, Cooper R. Roslund, Angela M. Crabtree, Peter B. Allen.

**Project administration:** Angela M. Crabtree.

**Supervision:** Paul A. Rowley.

**Writing – original draft:** Paul A. Rowley.

**Writing – review & editing:** Lance R. Fredericks, Mark D. Lee, Cooper R. Roslund, Peter B. Allen, Paul A. Rowley.

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
