## [Decision Letter · Decision Letter 0]

10 May 2020

PONE-D-20-06694

The design and implementation of restraint devices for the injection of pathogenic microorganisms into Galleria mellonella.

PLOS ONE

Dear Dr. Rowley,

Thank you for submitting your manuscript to PLOS ONE. After careful consideration, we feel that it has merit but does not fully meet PLOS ONE’s publication criteria as it currently stands. Therefore, we invite you to submit a revised version of the manuscript that addresses the points raised during the review process.

Three experts in this field reviewed your paper and found that this Galleria mellonella injection method presented by this study could be very useful for those working with this model organism. Having said that, the reviewer 1 and 3 raised several critical concerns that you need to pay attention to. In particular, the comparative data among suggested techniques appear to be critical to me. Other comments should also be addressed point-by-point. 

We would appreciate receiving your revised manuscript by Jun 24 2020 11:59PM. To enhance the reproducibility of your results, we recommend that if applicable you deposit your laboratory protocols in protocols.io, where a protocol can be assigned its own identifier (DOI) such that it can be cited independently in the future. For instructions see: http://journals.plos.org/plosone/s/submission-guidelines#loc-laboratory-protocols

We look forward to receiving your revised manuscript.

Kind regards,

Yong-Sun Bahn, Ph.D.

Academic Editor

PLOS ONE

2. Please amend either the abstract on the online submission form (via Edit Submission) or the abstract in the manuscript so that they are identical.

Reviewers' comments:

Reviewer's Responses to Questions

**Comments to the Author**

1. Is the manuscript technically sound, and do the data support the conclusions?

Reviewer #1: Partly

Reviewer #2: Yes

Reviewer #3: Yes

2. Has the statistical analysis been performed appropriately and rigorously? 

Reviewer #1: N/A

Reviewer #2: Yes

Reviewer #3: Yes

3. Have the authors made all data underlying the findings in their manuscript fully available?

Reviewer #1: Yes

Reviewer #2: Yes

Reviewer #3: Yes

4. Is the manuscript presented in an intelligible fashion and written in standard English?

Reviewer #1: Yes

Reviewer #2: Yes

Reviewer #3: Yes

5. Review Comments to the Author

Reviewer #1: This manuscript provides the community working with Galleria mellonella larvae with 2 alternative ways of larval injection which aim to reduce, or even fully prevent needle injuries for the person injecting larvae, and as a consequence, transmission of pathogens/microbes to the injecting person.

Overall I agree with the authors that both tools, the modified pippette tips and the plexi-glass chamber, are useful Tools to facilitate Galleria injection and reduce the potential danger of needle injuries. An Information /method very valuable to the community.

Regardless of the Tools shown here, I have some comments I d like the authors to address:

Major:

In my opinion, this manuscript is solely a description of a method/in fact a tool to facilitate a method already known to the community - nevertheless the article is structures like a Research paper in Methods/Results and Discussion, this might be due to the outline the journal recommends, but what has been shown here as "results" is basically a Repetition of the method part, so I suggest a re-structuring of the manuscript

The actual Galleria-survivial Assay Shows, that Candida glabrata kills larvae, injected with the help of one tool - but there has not been carried out a comparision between the 2 Tools (pippette tip or plexiglass chamber), nor to a method without any of These Tools. I therefore strongly encourage the authors to Show comparative data on this - with only one inocolum size.

Also, it is important to include: how many larvae per sample (per inocolum) have been infected, do the survival curves represent one Experiment? or an average survival of 3 Experiments - has the Assay been repeated at all? And at what temperature have the larvae been incubated.

As discussion Point the authors might add that many Groups use Insuline syringes to inject larvae - could they also be used? or is their needle to short to reach the larvae trapped in the two different Chambers

Minor:

line 83: "In this study...should start as a new Paragraph

line 87: I dont fully agree with the conclusion "significantly resuces the Manual handling"...as to me urging a larva into one of the Chambers could be similarly stressful to the animal as Holding it between the fingers? and the authors do not Show comparative data that proof this conclusion - the only obvious Advantage of both chamber is - and that is important - that piercing yourself with a needle is almost impossible

line 124 ff: many Groups have described that they inject larvae through one of the prolegs - because this was explained to cause least harm to the larvae (Fallon et al, Methods MolBiol 2012) - this would not be possible to do with larvae in a chamber - please comment on this - did the authors do comparative assays to compare if injecting via the cuticle is more harmful thatn through the opening of the prolegs?

line 146: define LT50

Reviewer #2: The paper tediously describes a novel technique of injection of restraint Galleria larvae which is the model for studies toxicology, immunology, pathology. The approach offered is advantageous as compared to the previous ones. The paper is well organized, the data are convincing and the substantial body of relevant literature is cited. There is a high probability that the article, if published, will be referenced in future research.

Reviewer #3: General comments:

The text submitted by Fredericks and colleagues is a technical paper focussed on reducing the risk of accidental inoculation of potentially harmful pathogens when using Galleria larvae as an in vivo model. Their solution is a set of ‘cheap and cheerful’ handling devices – described herein. The text is mostly, clearly written if a little superficial at times. The take-home message is straightforward but clarity is needed in a couple of areas.

Overall, this paper follows a very similar approach to another manuscript by Dalton et al. (2017) – albeit their designs are more elegant.

Dalton, J. P., Uy, B., Swift, S., & Wiles, S. (2017). A novel restraint device for injection of Galleria mellonella larvae that minimizes the risk of accidental operator needle stick injury. Frontiers in cellular and infection microbiology, 7, 99.

There are many administration techniques used for Galleria, and these can differ according to lab, e.g., chilling on ice prior to inoculation, using a pipette tip taped to the bench where the insect larva is held against (to avoid needlestick injury), brief submersion in ‘baths’ containing entomopathogens, etc.

This reviewer does agree that more standardisation for Galleria work is needed (having worked on these insects since 2008). However, some of the arguments used for suggesting these devices are better than others available are askew.

When referring to Dalton et al. (2017), the authors are critical of the need for multiple de-contamination rounds. The Galleria grabber, which is comprised of a large (15 mm) thick sponge and bulldog clip, is perhaps more environmentally friendly than the new devices explained herein, where there is a sizeable volume of single-use plastic.

Lines 52-55. Submersion is also quite common with biopesticide work. It would be prudent to include a reference here, perhaps a recent paper comparing gavage and direct injection e.g., Coates et al. (2019).

Coates, C. J., Lim, J., Harman, K., Rowley, A. F., Griffiths, D. J., Emery, H., & Layton, W. (2019). The insect, Galleria mellonella, is a compatible model for evaluating the toxicology of okadaic acid. Cell biology and toxicology, 35(3), 219-232.

Lines 70-72: this may be through, but intrahaemocoelic injection can be achieved using automated micro-injection platforms.

Lines 80 - 83: see comments above on the perceived betterment of the proposed designs versus that of Dalton et al (2017).

Line 87 (and elsewhere): It is unclear how ‘injection speed’ was measured

Lines 93-99: Unless this protocol has been published previously the authors must either cite the original text, or, provide specific details, e.g., centrifugation (speed, duration, temperature), PBS, pH? (filter-sterilised?), how many yeast were injected per larva?

Why did you culture C. glabrata at room temperature (~20oC) when it is a microbe usually found in human mucosa? Additionally, you then inoculate/incubate (Line 108) the insects at 37 degrees C. Consistency is needed.

There appears to be only one strain/species used – contrary to the title of the section.

Line 109: replace larva with ‘larvae’, and are with ‘were’

Line 144: please include the sample sizes, and number of technical versus biological replicates. Did the authors follow the ARRIVE guidelines regarding sample size and power calculations?

Line 157: re-usable acrylic sheets will surely need to be de-contaminated, as with the sponge-based device developed by Dalton et al (2017), it can act as a fomite (refer back to critical comments, lines 80-83).

Lines 160 – 163:

Although essential to the argument being made, the authors do not describe their protocol in the methods section, or the sample size used to generate this information.

Lines 163- 165: how are the authors determining ‘faster rate of injection’, qualitatively, arbitrary benchmark?

Line 203: the so-called ‘white solids’ represent insect fat body

Lines 221: provide specific detail to evidence this statement.

6. PLOS authors have the option to publish the peer review history of their article (what does this mean?). If published, this will include your full peer review and any attached files.

Reviewer #1: No

Reviewer #2: No

Reviewer #3: No

---

## [Author Response · Author response to Decision Letter 0]

25 Jun 2020

Response to reviewers’ comments

Reviewer #1:

 This manuscript provides the community working with Galleria mellonella larvae with 2 alternative ways of larval injection which aim to reduce, or even fully prevent needle injuries for the person injecting larvae, and as a consequence, transmission of pathogens/microbes to the injecting person.

Overall I agree with the authors that both tools, the modified pippette tips and the plexi-glass chamber, are useful Tools to facilitate Galleria injection and reduce the potential danger of needle injuries. An Information /method very valuable to the community.

Regardless of the Tools shown here, I have some comments I d like the authors to address:

Major:

In my opinion, this manuscript is solely a description of a method/in fact a tool to facilitate a method already known to the community - nevertheless the article is structures like a Research paper in Methods/Results and Discussion, this might be due to the outline the journal recommends, but what has been shown here as "results" is basically a Repetition of the method part, so I suggest a re-structuring of the manuscript

Manuscript organization revised: We have now removed a large section from the Materials and Methods section detailing the injection process and incorporated it into the Results/Discussion section. The Materials and Methods section is now only used to describe details relating to the more granular details of our study that are not mentioned in the Results/Discussion section (i.e. strains used, details of chamber construction, and data analysis). We have also added three new subheadings in the Results/Discussion section to compare different methods of injection, to describe the loading of our restraint devices, and the injection process using our devices. This new organization avoids the somewhat redundant description of the injection process that was written in the initial submission.

The actual Galleria-survivial Assay Shows, that Candida glabrata kills larvae, injected with the help of one tool - but there has not been carried out a comparision between the 2 Tools (pippette tip or plexiglass chamber), nor to a method without any of These Tools. I therefore strongly encourage the authors to Show comparative data on this - with only one inocolum size.

DONE: New data included for better comparison between the methods: Figure 4, tables 1 and 2 are now included and comparisons between the methods are made in the text at lines: 216-228. We have added additional data that directly compares the performance of the two injection chambers using the opportunistic pathogen C. glabrata. We did not compare the injection using finger and thumb because of the risk of laboratory acquired infection. We did however, compare the restrain devices to manual restraint between finger and thumb for injections of PBS. 

Also, it is important to include: how many larvae per sample (per inocolum) have been infected, do the survival curves represent one Experiment? or an average survival of 3 Experiments - has the Assay been repeated at all? And at what temperature have the larvae been incubated.

Additional details have been provided. We now more clearly detail the numbers of larvae used in each experiment in Table 1 and Table 2. 

As discussion Point the authors might add that many Groups use Insuline syringes to inject larvae - could they also be used? or is their needle to short to reach the larvae trapped in the two different Chambers

Added: Line 134 “We also expect that this method is fully compatible with the use of insulin syringes with shorter needles as prolegs are almost always positioned close to the opening in the injection chambers”

Minor:

line 83: "In this study...should start as a new Paragraph

Done.

line 87: I dont fully agree with the conclusion "significantly resuces the Manual handling"...as to me urging a larva into one of the Chambers could be similarly stressful to the animal as Holding it between the fingers? and the authors do not Show comparative data that proof this conclusion - the only obvious Advantage of both chamber is - and that is important - that piercing yourself with a needle is almost impossible

Done. We have removed this statement based on the reviewers concerns. The sentence now reads: (line 91) “Both devices provide increased protection of the operator from accidental needlestick injury and laboratory-acquired infection.”

line 124 ff: many Groups have described that they inject larvae through one of the prolegs - because this was explained to cause least harm to the larvae (Fallon et al, Methods MolBiol 2012) - this would not be possible to do with larvae in a chamber - please comment on this - did the authors do comparative assays to compare if injecting via the cuticle is more harmful thatn through the opening of the prolegs?

Comment: We describe the targeting of the proleg in the paper (lines 199-201) and demonstrate the technique in the supplementary movie S1. In our method the angle of injection is quite shallow because of the restraint of the larva in the injection chamber.

line 146: define LT50

Done: We have now more completely defined LT50. Line 221: “...with an average time to 50% lethality (LT50)...”

Reviewer #2:

The paper tediously describes a novel technique of injection of restraint Galleria larvae which is the model for studies toxicology, immunology, pathology. The approach offered is advantageous as compared to the previous ones. The paper is well organized, the data are convincing and the substantial body of relevant literature is cited. There is a high probability that the article, if published, will be referenced in future research.

Reviewer #3:

General comments:

The text submitted by Fredericks and colleagues is a technical paper focussed on reducing the risk of accidental inoculation of potentially harmful pathogens when using Galleria larvae as an in vivo model. Their solution is a set of ‘cheap and cheerful’ handling devices – described herein. The text is mostly, clearly written if a little superficial at times. The take-home message is straightforward but clarity is needed in a couple of areas.

Overall, this paper follows a very similar approach to another manuscript by Dalton et al. (2017) – albeit their designs are more elegant.

Dalton, J. P., Uy, B., Swift, S., & Wiles, S. (2017). A novel restraint device for injection of Galleria mellonella larvae that minimizes the risk of accidental operator needle stick injury. Frontiers in cellular and infection microbiology, 7, 99.

There are many administration techniques used for Galleria, and these can differ according to lab, e.g., chilling on ice prior to inoculation, using a pipette tip taped to the bench where the insect larva is held against (to avoid needlestick injury), brief submersion in ‘baths’ containing entomopathogens, etc.

This reviewer does agree that more standardisation for Galleria work is needed (having worked on these insects since 2008). However, some of the arguments used for suggesting these devices are better than others available are askew.

When referring to Dalton et al. (2017), the authors are critical of the need for multiple de-contamination rounds. The Galleria grabber, which is comprised of a large (15 mm) thick sponge and bulldog clip, is perhaps more environmentally friendly than the new devices explained herein, where there is a sizeable volume of single-use plastic.

Done: We have now clarified that both of our devices are fully reusable to address the environmental concerns of the reviewer. Lines 253 “The devices that we have described to restrain G. mellonella larvae are inexpensive, non-porous, and can be reused multiple times after sterilization or decontamination.”

Lines 52-55. Submersion is also quite common with biopesticide work. It would be prudent to include a reference here, perhaps a recent paper comparing gavage and direct injection e.g., Coates et al. (2019).

Done: Our We now included additional references that demonstrate the various methods of introducing pathogens and chemicals into G. mellonella (including the reference suggested by the reviewer) (Line 55)

Lines 70-72: this may be through, but intrahaemocoelic injection can be achieved using automated micro-injection platforms.

Comment: The authors are aware of autoinjection platforms for C. elegans and Zebrafish embryos, but not for Galleria. If the reviewer could provide us with a specific description and reference we would be happy to include it in the paper.

Lines 80 - 83: see comments above on the perceived betterment of the proposed designs versus that of Dalton et al (2017).

Done: We have reworded this section to better describe the Galleria grabber. We focus our evaluation of the device on its porosity, which is of primary concern when dealing with pathogenic microbes that require biological containment. Lines 79-84: “For the injection of G. mellonella, there is also a device named the “Galleria grabber” that has been developed for the restraint of larvae between layers of a sponge [34]. This method enables injection without the need for grasping larvae between finger and thumb and offers the user protection from accidental needlestick injury. However, the use of a porous sponge for multiple injections increases the chance of its contamination by pathogenic microbes, which presents a challenge for effective decontamination.” 

Line 87 (and elsewhere): It is unclear how ‘injection speed’ was measured

Done: We have rearranged and added to the description of the injection process to better describe the process that was timed for each injection. Lines 204-214 “The injection time with a single channel Hamilton syringe takes an average of 62 seconds per larva (SD +/- 12 seconds, n = 11) for the micropipette device, which includes loading the syringe, injection, larval release, and decontamination of the needle and syringe. Using a repeat dispenser increases the rate of injection to 8 seconds per larva for both the micropipette tip device (SD +/- 2 seconds, n = 30) and the acrylic glass device (SD +/- 3 seconds, n = 30). This is compared to 30 seconds (SD +/- 16 seconds, n = 30) to restrain and inject larvae between finger and thumb, using a needle resistant glove.”

Lines 93-99: Unless this protocol has been published previously the authors must either cite the original text, or, provide specific details, e.g., centrifugation (speed, duration, temperature), PBS, pH? (filter-sterilised?), how many yeast were injected per larva?

Done: Lines102-105 “Hemocytometer counts of these cultures were used to determine the number of yeasts used for each injection (8.0 × 105, 3.0 × 106, 4.6 × 106, and 5.0 × 106 C. glabrata cells per injection). Prior to injection, yeast cells were harvested by centrifugation at 8,000 × g for 1 min (25°C) and suspended in filter sterilized PBS (pH 7).

Why did you culture C. glabrata at room temperature (~20oC) when it is a microbe usually found in human mucosa? Additionally, you then inoculate/incubate (Line 108) the insects at 37 degrees C. Consistency is needed.

Comment: Initial growth at room temperature was chosen to prevent overgrowth of C. glabrata cultures to enable more consistent results and to accommodate the schedules of undergraduate researchers that spearheaded this project. In addition, C. glabrata is also found in the environment in diverse habitats, not just within the mammalian digestive tract. Growth at 37oC initiates rapid expression of virulence genes (such as adhesins) that enable host invasion.

There appears to be only one strain/species used – contrary to the title of the section.

 Done: Title changed to “Culturing and preparation of yeast cells for injection.”

Line 109: replace larva with ‘larvae’, and are with ‘were’

Done 

Line 144: please include the sample sizes, and number of technical versus biological replicates. Did the authors follow the ARRIVE guidelines regarding sample size and power calculations?

Done: We now include a new table (Table 1) that states the number of larvae used in each experiment. Figure legends for Fig 4 and Fig 5 describe the number of technical vs biological replicate for each experiment.

In line with the ARIVE guidelines, we also added details of the power analysis that we performed. Line 142 “A power analysis was performed to assess the required sample size using G*Power (V3.1) (One-tailed T-Test, � = 0.05, � = 0.2, effect size = 0.8). 

Line 157: re-usable acrylic sheets will surely need to be de-contaminated, as with the sponge-based device developed by Dalton et al (2017), it can act as a fomite (refer back to critical comments, lines 80-83).

 Done: see previous comment above.

Lines 160 – 163:

Although essential to the argument being made, the authors do not describe their protocol in the methods section, or the sample size used to generate this information.

Comment: based on other reviewers comments that this is a methods paper we have fully described all materials used and the details of larval handling etc. in the methods and described the method fully in the results section.

Lines 163- 165: how are the authors determining ‘faster rate of injection’, qualitatively, arbitrary benchmark?

Done: Rephrased the sentence Line 162 “This long occupancy time allows the loading of multiple larvae before injection and allows easier injection due to their predictable positioning within the device (Fig 1 and 2).”

Line 203: the so-called ‘white solids’ represent insect fat body

 Sentence revised – see next comment.

Lines 221: provide specific detail to evidence this statement. “In the rare case that an injected larva exudes excessive amounts of hemolymph or any white solids (from the fat body), then the individual is discarded because of the increased likelihood of injection-induced injury.”

Done: We have removed this sentence because of a lack of data. We initially included it to indicate that we sometimes have off-target injections that we have observed cause injury to larvae as described previously (Fallon et al, Methods Mol Biol (2012)). However, this user error is infrequent so we feel it is not a major concern (less than 1 in 1000 injections).

---

## [Decision Letter · Decision Letter 1]

15 Jul 2020

The design and implementation of restraint devices for the injection of pathogenic microorganisms into Galleria mellonella.

PONE-D-20-06694R1

Dear Dr. Rowley,

We’re pleased to inform you that your manuscript has been judged scientifically suitable for publication and will be formally accepted for publication once it meets all outstanding technical requirements.

Kind regards,

Yong-Sun Bahn, Ph.D.

Academic Editor

PLOS ONE

Additional Editor Comments (optional):

Reviewers' comments:

Reviewer's Responses to Questions

**Comments to the Author**

1. If the authors have adequately addressed your comments raised in a previous round of review and you feel that this manuscript is now acceptable for publication, you may indicate that here to bypass the “Comments to the Author” section, enter your conflict of interest statement in the “Confidential to Editor” section, and submit your "Accept" recommendation.

Reviewer #1: All comments have been addressed

Reviewer #3: All comments have been addressed

2. Is the manuscript technically sound, and do the data support the conclusions?

Reviewer #1: Yes

Reviewer #3: Yes

3. Has the statistical analysis been performed appropriately and rigorously? 

Reviewer #1: Yes

Reviewer #3: Yes

4. Have the authors made all data underlying the findings in their manuscript fully available?

Reviewer #1: Yes

Reviewer #3: Yes

5. Is the manuscript presented in an intelligible fashion and written in standard English?

Reviewer #1: Yes

Reviewer #3: Yes

6. Review Comments to the Author

Reviewer #1: the authors have responded to all comments and made adequate changes to the manuscript, I therefore suggest to accept the new Version of the manuscript

Reviewer #3: The authors have addressed all the concerns raised by reviewers and their manuscript is much improved.

7. PLOS authors have the option to publish the peer review history of their article (what does this mean?). If published, this will include your full peer review and any attached files.

Reviewer #1: No

Reviewer #3: No

---

## [Editor Report · Acceptance letter]

20 Jul 2020

PONE-D-20-06694R1 

The design and implementation of restraint devices for the injection of pathogenic microorganisms into *Galleria mellonella*

Dear Dr. Rowley:

I'm pleased to inform you that your manuscript has been deemed suitable for publication in PLOS ONE. Congratulations! Your manuscript is now with our production department. 

Kind regards, 

on behalf of

Dr. Yong-Sun Bahn 

Academic Editor

PLOS ONE